# P2X Purinergic Receptors Are Multisensory Detectors for Micro-Environmental Stimuli That Control Migration of Tumoral Endothelium

**DOI:** 10.3390/cancers14112743

**Published:** 2022-05-31

**Authors:** Giorgia Scarpellino, Tullio Genova, Elisa Quarta, Carla Distasi, Marianna Dionisi, Alessandra Fiorio Pla, Luca Munaron

**Affiliations:** 1Department of Life Sciences & Systems Biology, University of Torino, 10123 Torino, Italy; giorgia.scarpellino@unito.it (G.S.); tullio.genova@unito.it (T.G.); elisa.quarta995@edu.unito.it (E.Q.); alessandra.fiorio@unito.it (A.F.P.); 2Department of Pharmaceutical Sciences, University of Piemonte Orientale, 28100 Novara, Italy; carla.distasi@uniupo.it (C.D.); marianna.dionisi@uniupo.it (M.D.)

**Keywords:** purinergic receptors, calcium signaling, cell migration, tumor-derived endothelial cells

## Abstract

**Simple Summary:**

Extracellular ATP is highly concentrated in tumor stroma. In this study, we investigated the effects of the synthetic ATP analog Benzoylbenzoyl-ATP, 2′(3′)-O-(4-Benzoylbenzoyl)adenosine 5′-triphosphate (BzATP), an agonist for P2X receptors, on tumor-derived endothelial cells (TEC) obtained from three different human tumors (breast, kidney and prostate carcinomas, respectively, BTEC, RTEC and PTEC). Treatment with high BzATP concentrations (100 µM) significantly reduced migration of all TEC types, resulting ineffective on human normal microvascular endothelium (HMEC); intriguingly, both the functional effect and associated calcium signals are sensitive to some key biological parameters of tumor stroma that include pH, Ca^2+^ and Zn^2+^. The lack of calcium signals selectively observed in PTEC, in which BzATP still retains its functional effect, suggests variability of intracellular signaling among TEC. These findings provide novel insights into the role of extracellular ATP as a multisensory regulator of migratory potential in tumoral endothelium.

**Abstract:**

The tumoral microenvironment often displays peculiar features, including accumulation of extracellular ATP, hypoxia, low pH-acidosis, as well as an imbalance in zinc (Zn^2+^) and calcium (Ca^2+^). We previously reported the ability of some purinergic agonists to exert an anti-migratory activity on tumor-derived human endothelial cells (TEC) only when applied at a high concentration. They also trigger calcium signals associated with release from intracellular stores and calcium entry from the external medium. Here, we provide evidence that high concentrations of BzATP (100 µM), a potent agonist of P2X receptors, decrease migration in TEC from different tumors, but not in normal microvascular ECs (HMEC). The same agonist evokes a calcium increase in TEC from the breast and kidney, as well as in HMEC, but not in TEC from the prostate, suggesting that the intracellular pathways responsible for the P2X-induced impairment of TEC migration could vary among different tumors. The calcium signal is mainly due to a long-lasting calcium entry from outside and is strictly dependent on the presence of the receptor occupancy. Low pH, as well as high extracellular Zn^2+^ and Ca^2+^, interfere with the response, a distinctive feature typically found in some P2X purinergic receptors. This study reveals that a BzATP-sensitive pathway impairs the migration of endothelial cells from different tumors through mechanisms finely tuned by environmental factors.

## 1. Introduction

Extracellular purines and pyrimidines, including ATP, ADP, UTP and UDP, play important roles in a variety of physiological and pathological processes [1,2]. Their biological activity is mediated by P2X and P2Y purinergic membrane receptors (P2XRs and P2YRs, respectively). In particular, the P2XR subfamily includes seven members (P2X1–7) of ligand-gated ion channels (ionotropic receptors) opened by extracellular ATP (eATP). P2XRs form homo- and hetero-trimeric assembly of subunits with intracellular amino and carboxyl termini, two transmembrane α-helices and a large extracellular loop, whose structural and functional details have been deeply investigated by crystallographic and molecular tools [3].

P2XR channels are functional hubs that act as membrane sensors for different extracellular stimuli in addition to eATP [4,5,6]. They are allosterically modulated by a variety of inorganic ions, including protons (similar structure as ASIC [7,8,9]), zinc (Zn^2+^) [10], calcium (Ca^2+^) [11] and other cations in the tissue microenvironment, as well as by hypoxia [12,13]. This feature is relevant to accomplishing their physiological and pathological roles.

The tumoral microenvironment often displays alterations in many chemical and mechanical parameters that affect the physiology of cancer-related cellular components such as cancer cells, tumor-associated fibroblasts and immune cells. Indeed, the stroma of solid cancers is characterized by an accumulation of eATP [2,14], low oxygen levels-hypoxia [15,16], low pH-acidosis [17,18,19,20], the unbalance of Zn^2+^ [21,22], Ca^2+^- hypercalcemia [23,24] and, at least in some particular tumors such as pancreatic ductal adeno-carcinoma (PDAC), mechanical alteration of the extracellular matrix resulting in increased stiffness [25,26]. The disordered composition and structure of the stroma is qualitatively and quantitatively variable among tissues and perturbs tumor growth and angiogenesis [27,28,29,30,31,32].

In addition to high-affinity P2XRs, high concentrations of eATP released by dying cells in the inflammatory environment and cancer cells can recruit low-sensitive P2X7Rs, widely distributed in healthy tissues and over-expressed in inflammation and cancer [1,33]. In a previous report, we showed that some members of P2 receptors, including P2X7 and P2Y11, impair the migratory potential of tumor-derived endothelial cells (TECs) triggered by high eATP typically found in the tumor microenvironment [13,14].

The canonical view of P2XRs is focused on their function as ion channels, often Ca^2+^ permeable. Nonetheless, unusual behaviors have been reported. A debated biophysical property of P2X7R relies on its dual activity as an ‘ion channel’ switching to a ‘macrochannel’ upon long-lasting stimulation with high eATP concentrations, which allows ATP itself to pass the membrane. Moreover, increasing evidence opens the possibility for P2X7R to promote a variety of cellular responses that are not ultimately associated with its conductive function and the related intracellular Ca^2+^ signaling [34,35,36,37]. Consistently, some P2X7 variants (including P2X7B and nfP2X7 isoforms) lack the pore-forming cytotoxic activity [38].

The pharmacological modulation of the P2X family is very complex, showing high variability among species and a high degree of overlapping sensitivity among its members. A further complication relies on the formation of hetero-multimeric complexes that could reveal peculiar biophysical and functional features. As an example, P2X4 and P2X7 show the highest sequence similarities among P2X and can form heteromers: they are co-expressed in the secretory epithelium and immune and inflammatory cells, thus contributing to inflammation and nociception [39,40]. In addition, they modulate high glucose and palmitate-induced endothelial cell activation and dysfunction [39,40].

Considered as a whole, the variable pattern of P2XRs expressed in the plasma membrane undergoes multiple regulations from the local environment and mediates their biological contribution.

Here, we show that the synthetic ATP analog Benzoylbenzoyl-ATP, 2′(3′)-O-(4-Benzoylbenzoyl)adenosine 5′-triphosphate (BzATP), an agonist for P2XRs [41,42], impairs migration of three different human TEC lines, but is ineffective on normal microvascular endothelium; our data indicate a marked sensitivity of purinergic-related activity to extracellular pH, Ca^2+^ and Zn^2+^, resembling the tumoral microenvironment.

## 2. Materials and Methods

### 2.1. Cell Cultures

Breast tumor-derived endothelial cells (BTEC), renal tumor-derived endothelial cells (RTEC) and prostate tumor-derived endothelial cells (PTEC) were isolated from human breast lobular-infiltrating carcinoma biopsy, human renal and prostate carcinoma, respectively, and characterized in the laboratory of Professor Benedetta Bussolati (Department of Molecular Biotechnology and Health Sciences, University of Torino, Italy) [43].

Human microvascular endothelial cells (HMEC) were purchased from ATCC. (ATCC, Manassas, VA, USA).

TEC and HMEC were grown in EndoGRO-MV-VEGF (Merck Millipore, Burlington, MA, USA) Complete Media Kit, as previously described [13,32].

### 2.2. Calcium Imaging and Experimental Protocols

Cells were seeded at a density of 5000 cells/cm^2^ on glass coverslips and left for 24 h in an incubator (37 °C; 5% CO_2_). For ratiometric [Ca^2+^]c measurements, cells were loaded with 2 μM Fura–2AM (Invitrogen, Carlsbad, CA, USA) for 30 min at 37 °C and fluorescence was acquired by Nikon Eclipse TE-2000S (Minato, Tokyo, Japan) inverted microscope and Metafluor Imaging System (Molecular Devices, Sunnyvale, CA, USA). The sample was then excited at 340 nm and 380 nm alternatively, and the [Ca^2+^]c was expressed as a ratio (R) of emitted fluorescence at 510 nm corresponding to two excitation wavelengths.

For each experiment, at least 20 regions of interest (ROIs) were selected, each corresponding to a single cell in the chosen image field, and images were acquired every 3 s. In order to limit noise, real-time back-subtraction was applied.

Calcium imaging analysis was performed by peak amplitude and area quantification using Clampfit 11.1 (Axon PClamp, Molecular Devices, San Jose, CA, USA) and GraphPad Prism 6 (GraphPad Software, Inc., La Jolla, CA, USA) for statistical analysis. Peak amplitude was calculated as the highest value of R within 300 s following the treatment. In the presence of extracellular Ca^2+^, the area underlying the Ca^2+^ influx during the sustained phase was evaluated at 300 s after the onset of the response. The total area under Ca^2+^ spikes in 0 Ca^2+^_out_ was measured using the Event Detection protocol in Clampfit 11.1. 

### 2.3. Migration Assays

In vitro wound healing culture-insert assay. Cells were seeded in EndoGRO 5% FCS on 12-well culture plates using 3-well silicone culture inserts (IBIDI^®^ GmbH, Planegg, Germany), with a density of 5 × 10^6^ cells/mL. Cells were maintained in an incubator (37 °C; 5% CO_2_) until confluence within the 70 µL chambers was reached. Cell monolayers were starved for 2 h in EndoGRO 0% FBS, and then the inserts were removed. A wash with PBS w/Ca^2+^ and w/Mg^2+^ solution was completed before adding the treatments in duplicate. EndoGRO 5% FBS w/oVEGF was used as positive control (CTR).

Images were acquired using a Nikon Eclipse Ti-E microscope with a 10× objective. Cells were kept at 37 °C and 5% CO_2_, and pictures were taken every 2 h using MetaMorph® Software (Molecular Devices, Sunnyvale, CA, USA) [43,44]. Migration was measured up to 12 h with MetaMorph® Software and expressed as percentage of migration [45,46]. At least 4 fields for each condition were analyzed in each independent experiment

Random Migration Assay. BTEC were seeded at a density of 4000 cells/cm^2^ on 12-well culture plates coated with 1% gelatin in EndoGRO 5% FCS. Cells were then washed with PBS w/Ca^2+^ and w/Mg^2+^ solution and treatments were added in duplicate. EndoGRO 5% FCS w/o VEGF was used as control (CTR). The results represent normalized values against the corresponding CTR obtained in each of the three independent experiments carried out for each experimental condition.

Experiments were performed using the same set-up described above but using a Nikon Plan 20X objective. Images were acquired for 10 h every 10 min using MetaMorph® Software. Image stacks were then analyzed with ImageJ 1.8.0, and at least 250 cells/condition were tracked. Migration rate (µm/min) was obtained by measuring the distance covered by cells between two subsequent time points after conversion of pixels to micrometers. At least five fields for each condition were analyzed in each independent experiment. The results represent normalized values against the corresponding CTR obtained in each of the three independent experiments carried out for each experimental condition.

### 2.4. Statistical Analysis

Data were analyzed with GraphPad Prism 6 (GraphPad Software, Inc., La Jolla, CA, USA). In order to check the normal distributions of each dataset, a preliminary Shapiro–Wilk test was performed and accordingly to that, and to the number of datasets to be compared, statistical analysis was performed by using the Student’s *t*-test (or the non-parametric Mann–Whitney test) or the one-way ANOVA (or the non-parametric Kruskal–Wallis test). A *p*-value < 0.05 was considered significant.

## 3. Results

### 3.1. High ATP and BzATP Concentrations Inhibit Migration of Endothelial Cells Obtained from Different Human Tumors (Breast, Renal and Prostate Carcinomas) but Is Ineffective on Normal Human Microvascular EC (HMEC)

The first set of experiments was conducted in order to verify and extend what was previously shown in BTEC to RTEC and PTEC. The application of 100 µM ATP significantly reduced migration in all three cell types; on the other hand, a lower concentration (1 µM) was ineffective (Figure 1A).

The application of 100 µM BzATP impaired wound healing in BTEC starting from 8 h, with a relative effect being even more marked at 16 h (Figure 1B). Conversely, the same treatment failed to reduce the migration of normal endothelium (HMEC) (Figure 1B). All the following experiments are shown at 12 h. The response of the other TEC tested (RTEC and PTEC) resembled that observed in BTEC (Figure 1C). Free random migration of BTEC was also reduced by 100 µM BzATP application for 10 h (Figure 1D).

Pre-incubation with 100 µM pyridoxalphosphate-6-azophenyl-2′,4′-disulfonic acid (PPADS), a broadly used non-selective P2X antagonist, prevented this effect on all the TEC types (Figure 1E–G). Similar results were obtained by treatment with Suramin in BTEC (Appendix A).

### 3.2. BzATP Promotes Different Calcium Signals on TEC and NEC

Acute application of 100 µM BzATP on BTEC and RTEC evoked long-lasting cytosolic calcium signals (Figure 2A). The same response was observed on HMEC to a much lesser extent in terms of peak amplitude, area and percentage of responding cells (Figure 2A–D). The Ca^2+^ increase was strictly dependent on the presence of the agonist, as revealed by the rapid recovery to the resting Ca^2+^ levels following agonist removal (Figure 2E).

Intriguingly, the same exposure to BzATP failed to induce any Ca^2+^ rise in functional PTEC (Figure 2A,F).

The Ca^2+^ response was significantly decreased in Ca^2+^-free extracellular medium (Figure 3A), as shown by area quantification (Figure 3A–D); these observations suggest a strong component of Ca^2+^ entry.

Preincubation of BTEC with Suramin or pyridoxalphosphate-6-azophenyl-2′,4′-disulfonic acid (PPADS) completely prevented the response to BzATP (Figure 3E). Taken together, these features are fully compatible with P2RXs involvement.

### 3.3. BzATP-Dependent Calcium Entry in BTEC and RTEC Is Sensitive to Low pH and High Extracellular Zinc and Calcium

A remarkable feature of P2XRs is their sensitivity to different environmental ions, including protons, Zn^2+^ and Ca^2+^. In particular, their increased concentration inhibits P2X7 pore activation. For this reason, we investigated the effects of the Ca^2+^ response to 100 µM BzATP.

Pre-incubation with an acidic medium (pH 6.4) strongly reduced the Ca^2+^ increase triggered by BzATP in both BTEC and RTEC (Figure 4A,E), as clearly shown by all the quantitative parameters (Figure 4B–D,F–H).

Similar results were obtained upon pre-incubation with extracellular 50 µM ZnCl_2_ (but not 10 µM) or 10 mM CaCl_2_ in BTEC (Figure 5A–C). In addition, acute application of 50 µM ZnCl_2_ abrogates the calcium response to BzATP in BTEC.

Accordingly, treatment with 50 µM ZnCl_2_ significantly prevented the antimigratory activity of BzATP, while 10 mM CaCl_2_ produced a smaller effect (Figure 5E,F).

## 4. Discussion

Vascular endothelium is a highly versatile and flexible system, able to react and adapt to specific tissue microenvironments. This feature contributes to the huge variability of endothelial cell types highlighted by several reports in both normal and altered conditions, with tumor-derived endothelium displaying divergent phenotypic and functional features compared to the healthy counterpart.

We previously reported the ability of some purinergic agonists, including high concentrations of adenosine, ADP and ATP, but not UTP, to exert an anti-migratory activity on TEC [13,30]. They also trigger Ca^2+^ signals associated with release from intracellular stores and entry from the external medium: thus, Ca^2+^ increase following purinergic stimulation is a suitable early readout of the intracellular response, even if the causal relationship between Ca^2+^-dependent events and the impairment of migration is still elusive and needs further investigation.

Several lines of evidence point to a crucial role of P2X purinergic receptors in this process [13]. All P2X forms are expressed in human vascular ECs from healthy tissues, as shown by immunofluorescence, pharmacological and electrophysiological approaches [1,44]. Most of them are functional Ca^2+^-permeable channels and share the ability to be modulated by a number of environmental factors, including hypoxia, pH and inorganic ions that are often altered in tumors.

For this reason, we decided to investigate the effect of the ATP analog, BzATP, a non-selective agonist for P2XRs. BzATP has been reported to act on various recombinant homomeric P2XRs (i.e., P2X1, P2X2, P2X3, P2X4, P2X5 and P2X7) [45] and on several heteromeric P2XR (1/2, 2/3, 4/7) [46], showing variable pharmacology and biophysical properties [40,45].

Here, we report that high BzATP concentrations reduce the migratory potential of all TEC used but not of normal microvascular ECs (HMEC).

The same stimulation evoked an acute cytosolic calcium increase in BTEC, RTEC and HMEC. The Ca^2+^ signal is long-lasting and strictly dependent on the presence of the agonist and receptor occupancy: this feature fits well with the involvement of slow-inactivating P2X ionotropic receptors.

The ability of low pH to abolish Ca^2+^ signals confers an acid-sensitivity to the purinergic signaling, a property that could be highly relevant in the tumor microenvironment usually associated with strong acidosis associated with hypoxia. Direct pH sensitivity is compatible with the recruitment of some P2XR isoforms; moreover, the depression of Ca^2+^ signals and the recovery of TEC migration by high Zn^2+^ concentration (50 µM) nicely agrees with the contribution of P2XRs, as well as the modulation by increased extracellular Ca^2+^ concentration [5,6,45].

Different P2XRs could account for the strong sensitivity to extracellular Zn^2+^, Ca^2+^ and low pH. In addition, heteromeric P2XR complexes could be expressed in TEC, with peculiar pharmacology, already found in normal vascular endothelial cells, including P2X4/7 [39,40]. In addition, we cannot exclude the potential recruitment of non-canonical forms of P2XR isolate in different preparations [38]. Some P2XRs expressed in TEC could transduce biological information through non-canonical calcium-independent pathways; among them, several forms of P2X7 proteins have been reported to lack ion conductivity working independently on their ion channel activity.

However, our data cannot rule out indirect interference of pH, Zn^2+^ and Ca^2+^ on other cellular targets.

In addition to the modulation by inorganic ions, the sensitivity to suramin and PPADS, together with the long-lasting receptor-dependent Ca^2+^ signals, clearly points to the involvement of P2X ionotropic receptors.

Although the apparent correlation between Ca^2+^-dependent events and the alteration of migratory potential in TEC treated with BzATP, their causal relationship remains unclear. Indeed, here, we show that BzATP fails to exert any detectable Ca^2+^ signals in PTEC but retains the ability to inhibit migration; in this particular cell type, the purinergic stimulation could be mediated by non-canonical functions of P2XRs, independently of their ion conductivity. This observation agrees with recent evidence provided by our lab showing that UTP triggers strong Ca^2+^ increases (both calcium release and Ca^2+^ entry) without any interference with BTEC migratory potential [47]. The intracellular pathways underlying the inhibition of TEC motility could vary among different tumors.

In conclusion, the evidence supports the hypothesis of a key role of P2X receptors in the selective control of TEC migration. This event is modulated by a number of key environmental factors usually found in tumor stroma.

The data provided in this paper are promising but deserve further investigation, particularly concerning two aspects. Firstly, the detailed definition of the molecular composition of the P2X complexes gathered at the TEC plasma membrane, their interactome and the intracellular machinery responsible for the regulation of endothelial migration in tumors. These mechanistic advancements are required for the set-up of new pharmacological approaches aimed at interfering with tumor vascularization. A second remarkable issue is related to the biological meaning of the antimigratory activity exerted by eATP accumulation in the broad context of in vivo vascular remodeling and angiogenesis.

## 5. Conclusions

High concentrations of the ATP analog, BzATP, a P2XR agonist, are anti-migratory for different human TECs (from breast, kidney and prostate carcinomas) but not for normal microvascular ECs.

The functional effect of BzATP is modulated by extracellular pH, Ca^2+^ and Zn^2+^ ions, all established hallmarks of tumoral stroma, together with abnormally high levels of eATP. While some features are shared by all TEC tested, intracellular signaling pathways are variable and deserve further mechanistic insight.

## Figures and Tables

**Figure 1 cancers-14-02743-f001:**
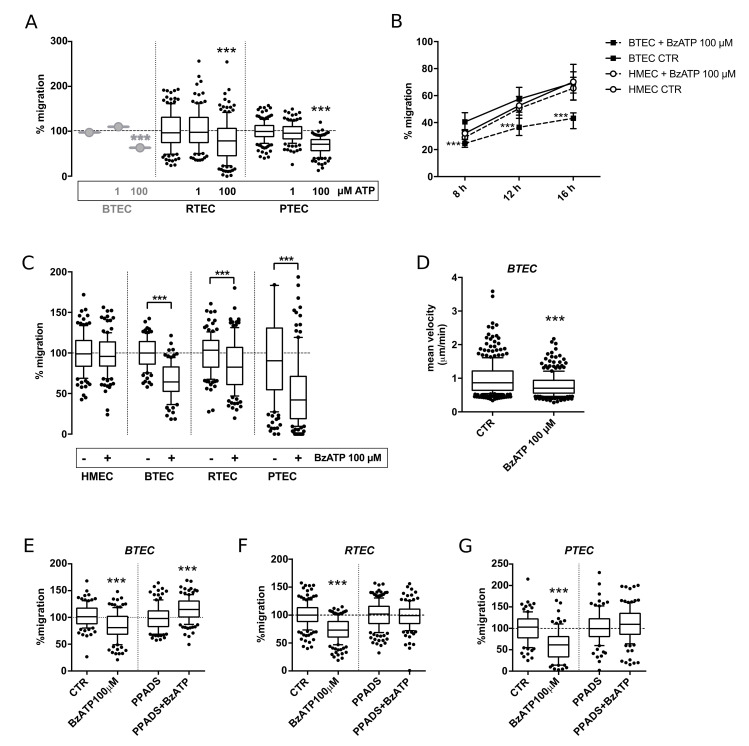
High concentrations of ATP or BzATP inhibited wound healing of TEC obtained from breast, renal and prostate carcinomas (respectively BTEC, RTEC and PTEC). (**A**) Effect of 1 µM and 100 µM ATP on wound healing in all 3 TEC types. (**B**) Representative wound healing experiments on BTEC and HMEC. Percentage of wound healing upon stimulation with 100 µM BzATP (8, 12, 16 h). Data are expressed as median and interquartile range for each condition (*n* ≥ 35). Student’s *t*-test: BzATP (dotted line) versus control at each time-point. *** *p*-value < 0.0005. (**C**) Percentage of TEC wound healing upon treatment with 100 µM BzATP (+) or in control condition (−) (12 h). Data are expressed as box and whiskers showing the median and 10–90 percentiles of 3 independent experiments for each condition (*n* ≥ 90), each normalized to the corresponding control. Student’s *t*-test or Mann–Whitney comparing BzATP to the corresponding control. *** *p*-value < 0.0005. (**D**) Single cell-free migration of BTEC treated or not with 100 µM BzATP (10 h). Mean velocity (µm/min) was evaluated from 3 independent experiments for each condition, each normalized to the corresponding control. Data are expressed as box and whiskers showing the median and 10–90 percentiles (*n* ≥ 200). Mann–Whitney test comparing BzATP treatment to control. *** *p*-value < 0.0005. (**E**–**G**) Percentage of BTEC wound healing upon treatment with 100 µM BzATP, 100 µM PPADS or BzATP and PPADS together in all the 3 TEC types (12 h). Data are expressed as box and whiskers showing the median and 10–90 percentiles of 3 independent experiments for each condition (*n* ≥ 90), each normalized to the corresponding control (CTR or PPADS). Mann–Whitney test comparing BzATP treatment to the corresponding control. *** *p*-value < 0.0005.

**Figure 2 cancers-14-02743-f002:**
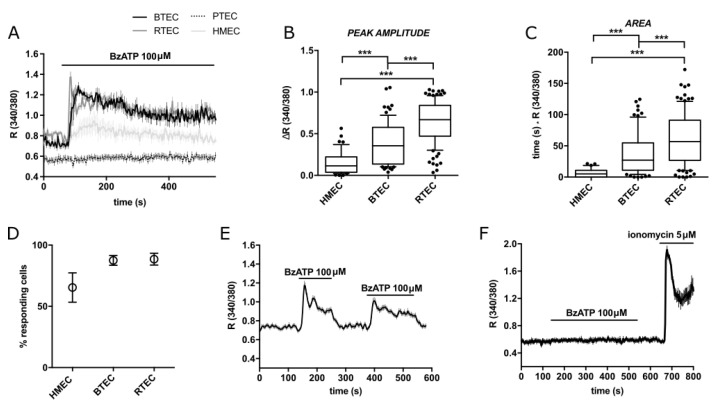
BzATP-triggered calcium signals in TEC and HMEC. (**A**) Representative traces of 100 µM BzATP-induced calcium signals in HMEC, BTEC, RTEC and PTEC. (**B**,**C**) Peak amplitude and area of calcium signals measured in (**A**) (see methods). Data are expressed as box and whiskers showing the median and 10–90 percentiles of at least 3 independent experiments for each condition (*n* ≥ 70). Kruskall–Wallis test: *** *p*-value < 0.0005. (**D**) Percentage of cells responsive to 100 µM BzATP out of total stimulated cells. Percentage was calculated on the total number of cells examined in a recording field. Data are expressed as mean value ± SEM of at least 3 independent experiments for each condition (*n* ≥ 3). (**E**) Representative average traces of 100 μM BzATP stimulation and washout in BTEC. (**F**) Representative traces of PTEC stimulation with the ionophore Ionomycin 5 µM, following the treatment with 100 µM BzATP, to check PTEC functionality. Traces in (**A**,**E**,**F**) show the mean value ± SEM of all cells in the recorded field of one representative experiment. At least 3 independent experiments were carried out for each condition (*n* ≥ 3).

**Figure 3 cancers-14-02743-f003:**
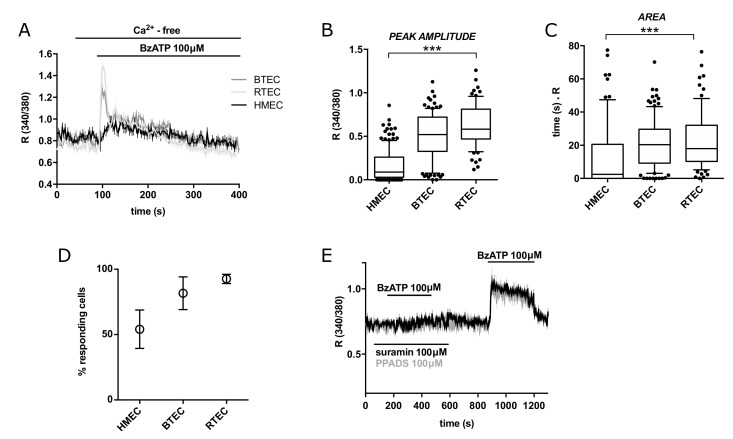
(**A**) Representative traces of 100 µM BzATP-induced calcium signals in HMEC, BTEC and RTEC in calcium-free extracellular solution (0 Ca_out_). (**B**,**C**) Peak amplitude and area of calcium signals measured in G. Data are expressed as box and whiskers showing the median and 10–90 percentiles of at least 3 independent experiments for each condition (*n* ≥ 75). Kruskall–Wallis test: *** *p*-value < 0.0005. (**D**) Percentage of cells responsive to 100 µM BzATP out of total stimulated cells. Percentage was calculated on the total number of cells examined in a recording field. Data are expressed as mean value ± SEM of at least 3 independent experiments for each condition (*n* ≥ 3). (**E**) Representative average traces of 100 μM BzATP stimulation in BTEC following the pre-incubation with 100 µM Suramin (black trace) or PPADS (grey trace) and after their washout. Traces in (**A**,**E**) show the mean value ± SEM of all cells in the recorded field of one representative experiment. At least 3 independent experiments were carried out for each condition (*n* ≥ 3).

**Figure 4 cancers-14-02743-f004:**
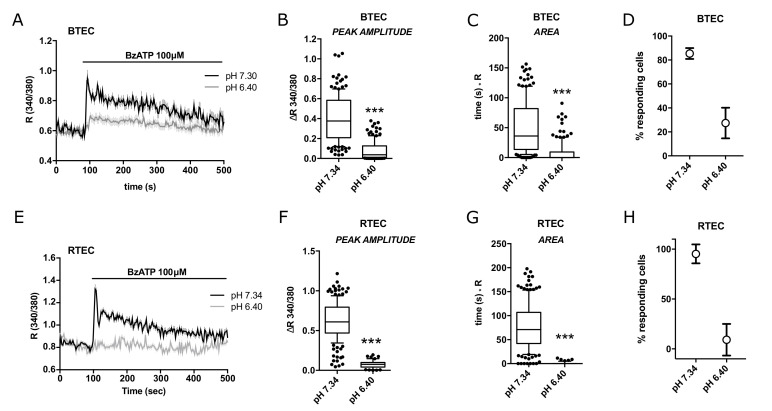
BzATP-triggered cytosolic calcium signals in BTEC and RTEC are modulated by extracellular pH. (**A**) Representative traces of 100 µM BzATP-induced calcium signals in BTEC and (**E**) RTEC measured in physiological extracellular solution (pH 7.4; black traces) and their modulation by acidic conditions (pH 6.4; grey traces). (**B**,**C**,**F**,**G**) Peak amplitude and area of calcium signals measured in (**A**,**E**), respectively. Data are expressed as box and whiskers showing the median and 10–90 percentiles of at least 3 independent experiments for each condition (*n* ≥ 50). Kruskall–Wallis test: *** *p*-value < 0.0005. (**D**,**H**) Percentage of cells responsive to 100 µM BzATP out of total stimulated cells in A and E, respectively. Percentage was calculated on the total number of cells examined in a recording field. Data are expressed as mean value ± SEM of at least 3 independent experiments for each condition (*n* ≥ 3). Traces in (**A**,**E**) show the mean value ± SEM of all cells in the recorded field of one representative experiment. At least 3 independent experiments were carried out for each condition (*n* ≥ 3).

**Figure 5 cancers-14-02743-f005:**
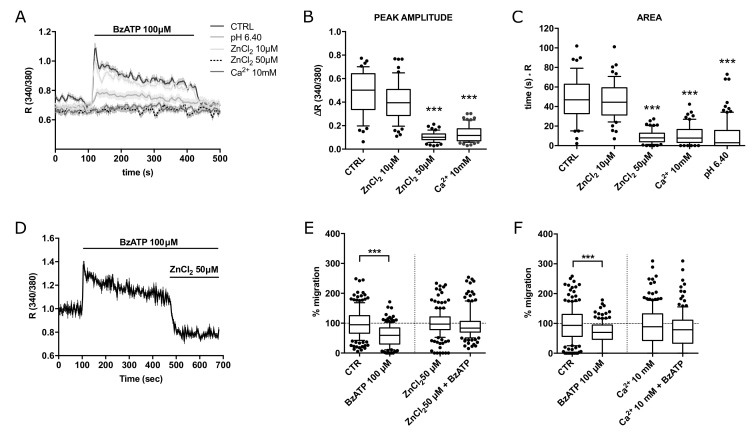
BzATP-dependent calcium signals in BTEC are sensitive to extracellular zinc and high calcium. (**A**) Representative traces of 100 µM BzATP-induced calcium signals in high zinc (10 and 50 µM) or calcium (10 mM) extracellular solution. (**B**,**C**) Peak amplitude and area of calcium signals shown in (**A**). Data are expressed as box and whiskers showing the median and 10–90 percentiles of at least 3 independent experiments for each condition (*n* ≥ 50). Kruskall–Wallis test: *** *p*-value < 0.0005. (**D**) Representative traces of the acute inhibitory effect of 50 µM ZnCl_2_ on BzATP-induced calcium signals. (**E**) Percentage of BTEC migration in wound healing experiments at 8 h treatment with 100 µM BzATP, 50 µM ZnCl_2_ or BzATP and 50 µM ZnCl_2_ together. Data are expressed as box and whiskers showing the median and 10–90 percentiles of 3 independent experiments for each condition (*n* ≥ 140), each normalized to the corresponding control (CTR or 50 µM ZnCl_2_). Mann–Whitney test comparing BzATP treatment to the corresponding control. *** *p*-value < 0.0005. (**F**) Percentage of BTEC migration in wound healing experiments at 8 h treatment with 100 µM BzATP, 10 mM CaCl_2_ or BzATP and 10 mM CaCl_2_ together. Data are expressed as box and whiskers showing the median and 10–90 percentiles of 3 independent experiments for each condition (*n* ≥ 140), each normalized to the corresponding control (CTR or 10 mM CaCl_2_). Mann–Whitney test comparing BzATP treatment to the corresponding control. *** *p*-value < 0.0005.

## Data Availability

Not applicable.

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
