# Peer review of "P2X Purinergic Receptors Are Multisensory Detectors for Micro-Environmental Stimuli That Control Migration of Tumoral Endothelium"

_cancers, 2022, doi:10.3390/cancers14112743_

Round 1

Reviewer 1 Report

My previous comments have been adequately addressed 

Reviewer 2 Report

The authors have addressed my suggestions.

Reviewer 3 Report

I am satisfied with the revised manuscript

This manuscript is a resubmission of an earlier submission. The following is a list of the peer review reports and author responses from that submission.

Round 1

Reviewer 1 Report

I appreciate the presentation of the data as box plots and the numbers of experiments clearly and appropriately described and appropriate methods

However I am unclear on the rigor of the conclusions drawn from the data. 

The authors state that P2X receptor expression and isoform variability can lead to the variable downstream outcomes observed in cells. They have previously shown that high ATP concentrations, as observed in a tumour environment, can activate P2X7Rs and inhibit the migration of endothelial cells extracted from breast cancer but not non-tumour derived cells. They have shown previously that this is dependent on cAMP but here they want to explore if this is due to P2XR evoked calcium signals, or non conductive effects. 

Firstly the authors explored the effect of high concentrations of BzATP on migration of endothelial cells extracted from different tumours. BzATP had an inhibitory effect on all tumour derived cells, however if this is to be linked to the specific activation of P2XRs, then the pharmacological (suramin and PPADS) inhibition needs to be shown on all the TECs. Moreover high and low levels of ATP should be used to show there is not an non specific effect from the BzATP. In their previous study the authors had  suggested high concentrations of  BzATP recruits the P2X7R and this played a role in the BzATP mediated inhibition of BTEC migration. So here the authors could have also use their siRNA approaches to specifically look at the role of P2X7.

BzATP evoked calcium signals in the MECs and TECs from breast and kidney but not from prostate. This suggests that the calcium signals are not necessary for the inhibition as BzATP does not inhibit MEC migration and does not evoke signals in prostate TECs. Here the authors go on to explore the effect of the tumour environment on the calcium signals but this section is without a clear conclusion. Demonstrate that BzATP evoked calcium responses are not necessary (especially as previously shown that BzATP inhibition of migration of TEC from breast is dependent on cAMP) or different tumour cells have different P2X-dependent signaling mechanisms.

Author Response

Reviewer 1

I appreciate the presentation of the data as box plots and the numbers of experiments clearly and appropriately described and appropriate methods

However I am unclear on the rigor of the conclusions drawn from the data. 

The authors state that P2X receptor expression and isoform variability can lead to the variable downstream outcomes observed in cells. They have previously shown that high ATP concentrations, as observed in a tumour environment, can activate P2X7Rs and inhibit the migration of endothelial cells extracted from breast cancer but not non-tumour derived cells. They have shown previously that this is dependent on cAMP but here they want to explore if this is due to P2XR evoked calcium signals, or non conductive effects. 

Firstly the authors explored the effect of high concentrations of BzATP on migration of endothelial cells extracted from different tumours. BzATP had an inhibitory effect on all tumour derived cells, however if this is to be linked to the specific activation of P2XRs, then the pharmacological (suramin and PPADS) inhibition needs to be shown on all the TECs. Moreover high and low levels of ATP should be used to show there is not an non specific effect from the BzATP. In their previous study the authors had suggested high concentrations of  BzATP recruits the P2X7R and this played a role in the BzATP mediated inhibition of BTEC migration. So here the authors could have also use their siRNA approaches to specifically look at the role of P2X7.

We fully agree with the reviewer. Accordingly, we added new experimental evidence in results section of the revised paper.

We tested the effects of P2XR-non-selective antagonist PPADS on RTEC and PTEC migration: as shown in revised figure 1, incubation with 100 µM PPADS completely prevented the anti-migratory activity of 100 µM BzATP in both RTEC and PTEC, similarly to what previously observed in BTEC.

Application of lower ATP concentration (1 mM) failed to exert any significant decrease on RTEC and PTEC migration, in line with BTEC. These data strengthen the hypothesis for the key role of low affinity P2XRs as membrane sensors in response to the abnormal accumulation of extracellular ATP detected in tumor stroma: however, the actual composition of the functional P2X (homomeric/heteromeric) complexes that cluster in TEC plasma membrane is a very complex issue that cannot be solved by the use of silencing or by pharmacological strategies due to the lack of reliable agonists/antagonists able to selectively distinguish among different P2X isoforms (see also our reply to reviewer 3 dealing with the broad issue). In order to investigate in more detail these and other questions, we are focusing on inducible models in which normal endothelial cells are conditioned by cancer cells as well as well as by a ‘tumor’ mimicking environment in controlled conditions (personal communication, paper in preparation). The main advantage of such a strategy relies on its more reliability and suitability: moreover, we can better control the effects of different components of tumor-released microenvironmental components on NEC-TEC transition. Preliminary results suggest the intriguing hypothesis that NEC conditioned with cancer cells remodel their P2X expression pattern and acquire the sensibility to BzATP typically observed only on TEC: the inductive potential could depend on the CC types and on the reciprocal paracrine crosstalk between EC and CC as well as on the microenvironmental parameters (mainly hypoxia and acidosis). We are collecting data on 2D co-cultures and 3D co-cultures with spheroids and exosome analysis: further investigation is needed to provide a detailed landscape of the purinergic-related machinery responsible for the regulation of endothelial migration remodelled by cancer cells through diffusible organic and inorganic (ions) mediators (personal communication, paper in preparation).

BzATP evoked calcium signals in the MECs and TECs from breast and kidney but not from prostate. This suggests that the calcium signals are not necessary for the inhibition as BzATP does not inhibit MEC migration and does not evoke signals in prostate TECs. Here the authors go on to explore the effect of the tumour environment on the calcium signals but this section is without a clear conclusion. Demonstrate that BzATP evoked calcium responses are not necessary (especially as previously shown that BzATP inhibition of migration of TEC from breast is dependent on cAMP) or different tumour cells have different P2X-dependent signaling

We thank the reviewer for this interesting concern focusing on a particularly intriguing issue.

The absence of detectable calcium signals in PTEC stimulated with 100 µM BzATP opens the possibility for the expression of ‘non-conductive’ forms of P2X that trigger intracellular signaling pathways independently on calcium fluxes: similar forms have been reported in some cancers (for a review, see Pegoraro A et al. 2021, see revised manuscript). In other models, such as hormone-refractory prostate cancer, cultured microglia and pancreatic cells, calcium independent intracellular signaling parhways were described even in the presence of functional calcium fluxes (Amstrup & Novak 2003; Shabbir M et al 2008; Morioka N et al 2008; Kopp R et al 2019; see the revised manuscript).

A possible explanation is that TEC migration could be under the control of intracellular machineries that differ in endothelial cell types: in some of them, calcium-dependent signals could be required, while in others they could act as a modulatory mechanisms or, like in the case of PEC, they are completely negligible. This is in nice agreement with the well described variability of normal and altered endothelium. In BTEC and RTEC, calcium signaling may work in an integrated complex framework with other intracellular signaling pathways to drive the overall reduction in migratory potential. Nonetheless, we cannot exclude that, even in the presence of calcium increase induced by purinergic stimulation (such as in BTEC and RTEC), this intracellular event could not be a major causal factor for the antimigratory effect. Our previous report seems to support this hypothesis (Scarpellino et al. Cancers 2019).

Further investigation on the detailed downstream pathways responsible for purinergic control of TEC migration and their variability would be helpful in order to selectively interfere with vascular remodeling in specific cancers.

This issue has been discussed in more detail in the revised version of the manuscript.

Reviewer 2 Report

This paper reports that the effects of the synthetic ATP analogue Benzoylbenzoyl-ATP, 2′(3′)-O-(4-Benzoylbenzoyl)adenosine 5′-triphosphate (BzATP), an agonist for P2X receptors, on tumor-derived endothelial cells (TEC) obtained from three different human tumors. Treatment with high BzATP concentrations (100 µM), significantly reduced migration of TEC, but was ineffective on human normal microvascular endothelium (HMEC). Moreover, the functional effect and the associated calcium signals are sensitive to some key biological parameters of tumor stroma that include pH, Ca2+ and Zn2+. Therefore, I believe that the issues on this study is worthy of investigation. However, the authors need to supplement some experimental data to further improve the scientific quality of the paper before acceptance of the manuscript. The main comments and recommendations are as follows.

  1. Your manuscript needs to be carefully edited and polished in terms of grammar and sentence structure before resubmission.
  2. The figure legends is not clear and needs to be modified. For example: in Figure 2E, The time period after agonist elution is not shown.
  3. There are some small mistakes in your manuscript layout that you need to check again. For example: Figgs 3A and figgs 3B.
  4. The data image layout is a bit messy, especially in Figure 2.
  5. This paper carried out some in vitro experiments on the cellular level of BzATP. How is the concentration of BzATP (100 µM) determined? For further research significance, you should continue to conduct subsequent animal studies.

Author Response

Reviewer  2

This paper reports that the effects of the synthetic ATP analogue Benzoylbenzoyl-ATP, 2′(3′)-O-(4-Benzoylbenzoyl)adenosine 5′-triphosphate (BzATP), an agonist for P2X receptors, on tumor-derived endothelial cells (TEC) obtained from three different human tumors. Treatment with high BzATP concentrations (100 µM), significantly reduced migration of TEC, but was ineffective on human normal microvascular endothelium (HMEC). Moreover, the functional effect and the associated calcium signals are sensitive to some key biological parameters of tumor stroma that include pH, Ca2+ and Zn2+. Therefore, I believe that the issues on this study is worthy of investigation. However, the authors need to supplement some experimental data to further improve the scientific quality of the paper before acceptance of the manuscript. The main comments and recommendations are as follows.

  1. Your manuscript needs to be carefully edited and polished in terms of grammar and sentence structure before resubmission.

The manuscript has been completely revised in grammar, sentence structure and typing.

  1. The figure legends is not clear and needs to be modified. For example: in Figure 2E, The time period after agonist elution is not shown.

Figure legends were revised and updated according to new evidence provided in the revised manuscript.

  1. There are some small mistakes in your manuscript layout that you need to check again. For example: Figgs 3A and figgs 3B.

Fixed.

  1. The data image layout is a bit messy, especially in Figure 2.

Fixed.

  1. This paper carried out some in vitro experiments on the cellular level of BzATP. How is the concentration of BzATP (100 µM) determined? For further research significance, you should continue to conduct subsequent animal studies.

We choose the concentration of 100 mM BzATP for continuity with our previous related paper (Avanzato et al. 2016) and according to the literature (see among the others: Kovacs et al, Cell Calcium 2018; Morioka et al, Glia 2008). This is the most powerful concentration in TEC, as revealed by dose-effect experiments (not shown).

We completely agree with the reviewer about the request for future investigation in vivo due to different reasons. In vitro studies allow to manage the environmental parameters in controlled conditions, but lack the complexity of the tissutal context: in particular, in the case of cancer microenvironment, the interaction with other cellular components of tumoral stroma is missing as well as the multi-faceted pattern of diffusible signaling compounds (growth factors, interleukins, hormones and others).

A sort of an ‘intermediate’ approach between in vitro canonical cell cultures and in vivo models could be the use of complex in vitro strategies that overcome some limitations of the 2D in vitro level, bringing the system closer to the in vivo one. We recently undertook this experimental setup an focused on inducible models in which normal endothelial cells are conditioned by cancer cells as well as well as by a ‘tumor’ mimicking environment in controlled conditions. The main advantage of such an approach relies on its more reliability and suitability: moreover, we can better control the effects of different components of tumor-released microenvironmental components on NEC-TEC transition. Preliminary results suggest the intriguing hypothesis that NEC conditioned with cancer cells remodel their P2X expression pattern and acquire the sensibility to BzATP typically observed only on TEC: the inductive potential could depend on the CC types and on the reciprocal paracrine crosstalk between EC and CC as well as on the microenvironmental parameters (mainly hypoxia and acidosis). We are collecting data on 2D co-cultures and 3D co-cultures with spheroids and exosome analysis: further investigation is needed to provide a detailed landscape of the purinergic-related machinery responsible for the regulation of endothelial migration remodelled by cancer cells through diffusible organic and inorganic (ions) mediators (personal communication, paper in preparation).

Reviewer 3 Report

The manuscript by Scarpellino et al investigates the effect of a synthetic ATP analogue on migration of tumour derived endothelial cells (TEC), without having significant effects on non-tumour endothelial cells (HMEC). 

The paper would benefit from additional experiments to determine which receptors are expressed and to what level on the different cell types used, this may help to unravel the different effects of the agonist seen on different cell types used here. This could then be followed up with over expression in HMEC (assuming they lack expression of one or more of the receptor family that is seen in the TEC) as a proof of principle experiment.

minor comments

please increase the font size on all figures - very hard to read!

the manuscript should be reviewed by a native English writer as there are a number of typographical and grammatical errors throughout

Author Response

Reviewer 3

The manuscript by Scarpellino et al investigates the effect of a synthetic ATP analogue on migration of tumour derived endothelial cells (TEC), without having significant effects on non-tumour endothelial cells (HMEC). 

The paper would benefit from additional experiments to determine which receptors are expressed and to what level on the different cell types used, this may help to unravel the different effects of the agonist seen on different cell types used here. This could then be followed up with over expression in HMEC (assuming they lack expression of one or more of the receptor family that is seen in the TEC) as a proof of principle experiment.

The reviewer rises a very relevant issue that concerns expression and functional pattern of P2X involved in the regulation of migration in NEC and TEC. Here we show that application of lower ATP concentration (1 mM) failed to exert any significant decrease on RTEC and PTEC migration, in line with BTEC. These data strengthen the hypothesis for the key role of low affinity P2XRs as membrane sensors in response to the abnormal accumulation of extracellular ATP detected in tumor stroma: however, the actual composition of the functional P2X (homomeric/heteromeric) complexes that cluster in TEC plasma membrane is a very complex issue that cannot be successfully elucidated by the simple use of silencing or by pharmacological strategies due to the lack of reliable agonists/antagonists able to selectively distinguish among different P2X isoforms (see Schmid & Evans, Ann Rev Physiol 2019; Gever et al, Pflugers Archiv 2019). In order to investigate in more detail these and other questions, we are focusing on inducible models in which normal endothelial cells are conditioned by cancer cells as well as well as by a ‘tumor’ mimicking environment in controlled conditions. The main advantage of such an approach relies on its more reliability and suitability: moreover, we can better control the effects of different components of tumor-released microenvironmental components on NEC-TEC transition. Preliminary results suggest the intriguing hypothesis that NEC conditioned with cancer cells remodel their P2X expression pattern and acquire the sensibility to BzATP typically observed only on TEC: the inductive potential could depend on the CC types and on the reciprocal paracrine crosstalk between EC and CC as well as on the microenvironmental parameters (mainly hypoxia and acidosis). We are collecting data on 2D co-cultures and 3D co-cultures with spheroids and exosome analysis: further investigation is needed to provide a detailed landscape of the purinergic-related machinery responsible for the regulation of endothelial migration remodelled by cancer cells through diffusible organic and inorganic (ions) mediators (personal communication, paper in preparation).

minor comments

please increase the font size on all figures - very hard to read!

Fixed

the manuscript should be reviewed by a native English writer as there are a number of typographical and grammatical errors throughout

The manuscript has been completely revised in grammar, sentence structure and typing.